# Rare Etiologies in Immune-Mediated Cerebellar Ataxias: Diagnostic Challenges

**DOI:** 10.3390/brainsci12091165

**Published:** 2022-08-30

**Authors:** Marios Hadjivassiliou, Mario Manto, Hiroshi Mitoma

**Affiliations:** 1Academic Department of Neurosciences, Royal Hallamshire Hospital, Glossop Road, Sheffield S10 2JF, UK; 2Service de Neurologie, Médiathèque Jean Jacquy, CHU-Charleroi, 6000 Charleroi, Belgium; 3Service des Neurosciences, University of Mons, 7000 Mons, Belgium; 4Department of Medical Education, Tokyo Medical University, Tokyo 160-0023, Japan

**Keywords:** cerebellum, immune-mediated cerebellar ataxias, autoimmune ataxia, primary autoimmune cerebellar ataxia

## Abstract

The cerebellum is particularly enriched in antigens and represents a vulnerable target to immune attacks. Immune-mediated cerebellar ataxias (IMCAs) have diverse etiologies, such as gluten ataxia (GA), post-infectious cerebellitis (PIC), Miller Fisher syndrome (MFS), paraneoplastic cerebellar degeneration (PCD), opsoclonus myoclonus syndrome (OMS), and anti-GAD ataxia. Apart from these well-established entities, cerebellar ataxia (CA) occurs also in association with autoimmunity against ion channels and related proteins, synaptic adhesion/organizing proteins, transmitter receptors, glial cells, as well as the brainstem antigens. Most of these conditions manifest diverse neurological clinical features, with CAs being one of the main clinical phenotypes. The term primary autoimmune cerebellar ataxia (PACA) refers to ataxic conditions suspected to be autoimmune even in the absence of specific well-characterized pathogenic antibody markers. We review advances in the field of IMCAs and propose a clinical approach for the understanding and diagnosis of IMCAs, focusing on rare etiologies which are likely underdiagnosed. The frontiers of PACA are discussed. The identification of rare immune ataxias is of importance since they are potentially treatable and may lead to a severe clinical syndrome in absence of early therapy.

## 1. Introduction

Immune-mediated pathologies frequently involve the cerebellum, leading to progressive ataxia characterized by dysmetria both in motor and cognitive domains [1,2,3,4,5]. Immune-mediated cerebellar ataxias (IMCAs) encompass diverse etiologies, including gluten ataxia (GA), post-infectious cerebellitis (PIC), Miller Fisher syndrome (MFS), paraneoplastic cerebellar degeneration (PCD), opsoclonus myoclonus syndrome (OMS), and anti-GAD ataxia. The triggers of autoimmunity are clear in GA (gluten sensitivity), PIC and MFS (infections), PCD (malignancy), and OMS (infections or malignant tumors). Most of these IMCAs are associated with well-characterized or partly characterized autoantibodies (Table 1) [2,6]. The clear presence of autoimmune triggers and the common clinical phenotype are important requirements for an independent disease entity, although autoimmune triggers remain unknown in anti-GAD ataxia.

Hadjivassiliou et al. (2017) [7] investigated the prevalence of IMCAs in 1500 UK patients with progressive ataxia. The authors reported that 30% of the patients had familial/genetic ataxia, although some did not have clear family history, and that 9% of the patients had cerebellar variant of multiple systemic atrophy (MSA-C) (prevalence out of total progressive ataxic cases). Apart from the above, 25% had definite IMCAs; 20% had GA, 2% had PCD, 2% had anti-GAD ataxia, 1% had PIC, and <1% had OMS. It should be acknowledged that 19% of the patients were classified as idiopathic sporadic ataxia. This category, approximately equivalent to the category of sporadic ataxia with adult-onset [8], probably included a large number of patients with IMCA [7,9].

Various autoimmune conditions with CAs have been reported recently. The recent progress in this field has enhanced our understanding of a group of not so well characterized IMCAs:

1. Cerebellar ataxias (CAs) are sometimes associated with autoantibodies known to have pathological effects, including antibodies (Abs) against ion channels and related proteins, synaptic adhesion/organizing proteins, and transmitter receptors. The actions of these Abs have been well-characterized by a series of studies on autoimmune limbic encephalitis [10,11,12,13]. In this regard, we proposed previously a nomenclature based on the pathogenic Abs, i.e., the pathogenic Ab to be coined to the name of the clinical subtype (e.g., anti-GAD ataxia) [9];

2. It has been reported that autoimmunity can target glial cells (e.g., autoimmune GFAP astrocytopathy) [14], and induces perivascular T-cell inflammation in the brainstem (e.g., CLIPPERS) [15], leading to the development of Cas;

3. Another concept was proposed recently for comprehensive understanding of IMCA. Primary autoimmune cerebellar ataxia (PACA) refers to those predominant CAs suspected as being autoimmune but lack any specific well-characterized pathogenic Ab markers [16,17].

The aim of this review is to clarify positions of recently reported subtypes within the IMCAs as well as the spectrum of PACA. Towards this goal, we describe the clinical features of these rare subtypes of IMCAs as well as those of PACA.

## 2. CA Associated with Autoantibodies against Ion Channels and Related Proteins

### 2.1. Anti-VGCC Ataxia

The P/Q-type voltage-gated Ca^2+^ channel (VGCC) is involved in the release of neurotransmitters at cerebellar neurons and motor nerve terminals. Anti-P/Q type VGCC Ab (abbreviated here as anti-VGCC Ab) is found in PCDs. A proportion of patients with anti-VGCC Ab and PCDs were reported to develop Lambert–Eaton myasthenic syndrome (LEMS) [18,19].

The most frequent cancer in patients with anti-VGCC Ab-positive PCDs is small cell lung cancer (SCLC) [20]. Based on a study of patients with PCDs and SCLC, anti-P/Q-type VGCC Ab was positive in seven of nine patients with PCDs and LEMS, and in 20% of patients with PCDs without LEMS [21]. Another study showed no difference in the clinical profiles between 16 anti-VGCC Ab-positive patients with PCDs and SCLC and nine anti-Hu Ab-positive patients with PCDs and SCLC [20]. Of these patients, one showed good responses to therapies, five showed stabilization at low Rankin scores, and five showed stabilization or worsening with high Rankin scores. The median survival time of these patients was 12 months. On the other hand, anti-VGCC Ab was also identified in patients free of paraneoplastic conditions. One study reported the presence of anti-VGCC Ab in 8 of 67 patients who showed chronic cerebellar degeneration [22]. Good prognosis was reported in these patients with non-paraneoplastic conditions. In both paraneoplastic and non-paraneoplastic conditions, immunotherapies were used, including intravenous immunoglobulin (IVIg), prednisone, and mycophenolate mofetil.

Liao and coworkers (2008) [23] provided evidence for the pathogenic actions of anti-VGCC Ab in the development of CA. Injection of a polyclonal peptide Ab against the major immunogenic region in the P/Q-type VGCCs (the extracellular domain-III S5-6 loop) impaired the functions of neuronal and recombinant P/Q-type VGCC, and consequently resulted in decrease in Ca^2+^ currents, decrease in glutamate, and manifestation of ataxic symptoms in mice.

### 2.2. Anti-Caspr2 Ataxia

Contactin-associated protein-like 2 (Caspr2) is an associated protein of voltage-gated K^+^ channel (VGKC) Kv1. Caspr2 forms a molecular complex with transient axonal glycoprotein-1 (TAG-1)/contactin-2, and VGKC Kv1 in compartments critical for neuronal activity and is required for proper positioning of Kv.1 [24]. Anti-Caspr2-associated neurological disease is characterized by diverse neurological symptoms. CA issometimes found with other neurological features or alone.

In a study of 38 patients with anti-Caspr2 Ab-associated encephalitis [25], 77% of the patients showed three or more core manifestations, including encephalic cerebral symptoms [cognitive disturbance (79%) and epilepsy (53%), CA (35%), insomnia (68%), neuropathic pain (61%), peripheral nerve hyperexcitability (54%), autonomic dysfunction (44%), and weight loss (58%)]. Neoplastic lesions were also detected in 19% of the patients (mostly thymoma) [25]. CSF examination was usually normal, although cell proliferation or a high protein level was observed in some patients [25]. MRI was in general normal [25]. About 50% of the patients showed full or good recovery after immunotherapy, including IVIg, corticosteroids, and plasma exchange, either alone or in combination, suggesting a good prognosis [25].

Another retrospective study identified stereotypicepisodes of paroxysmal CA in 5 of 20 patients [26]. The ataxic episodes, including gait imbalance, limb ataxia, and slurred speech, usually lasted a few minutes to a few days and usually improved following immunotherapy. The authors proposed the addition of paroxysmal CA to the spectrum of anti-Caspr2 Ab-associated neurological syndrome [26].

Interestingly, the heterogeneous nature of the anti-Caspr2 Ab-associated syndrome was confirmed in a subsequent study [27]. In general, permanent cerebellar ataxias and episodic ataxias are not associated with neuromyotonia or Morvan syndrome but with limbic encephalitis [27]. Thus, anti-Caspr2 can be classified into two groups: the limbic predominant group and peripheral nerve hyperexcitability-predominant (PNH) group, with no overlap [27]. Patients of the former group (52%) developed limbic symptoms alone (LE/−) or with extra-limbic symptoms (LE/+). Permanent CA was observed in 73% of the LE/+ patients. Episodic CA was found in 55% of these patients who showed LE/+ (LE and extra-limbic symptoms). These patients with LE showed Human Lymphocyte Antigen (HLA)-DRB1 *11:01, had high serum titers of anti-Caspr2 Ab, and were positive for anti-Caspr2 Ab in CSF. On the other hand, the latter group (48%) developed mild PNH alone or in combination with weight loss, hyperkinetic movement, dysautonomia, and agrypnia excitata resembling Morvan syndrome. Patients with PNH commonly showed no HLA association, low serum titers of anti-Caspr2 Ab, absent anti-Caspr2 Ab in CSF, and association with malignant thymoma.

A study using a cell-based study reported that anti-Caspr2 Ab from the patients with LE and PNH did not induce internalization of Caspr2, but rather interfered with the binding of Caspr2 to TAG-1/contactin-2 in hippocampal neuron cultures, which resulted in interference with the clustering of juxtaparanodes and in hyperexcitability [28].

### 2.3. Anti-DPPX Ataxia

Dipeptidyl-peptidase-like protein-6 (DPPX, DPP6), a regulatory subunit of VGKC Kv4.2, is a target antigen in autoimmune limbic encephalitis. The autoimmune limbic encephalitis with anti-DPPX Ab exhibits divergent neurological symptoms. CA is either the sole clinical manifestation or appears in conjunction with other neurological symptoms. A very prominent and characteristic feature of this entity is the dramatic loss of weight that may precede the neurological presentation.

One systematic study reported the multifocal neurological symptoms in 20 patients [29]. Of the 20 patients, 12 were men with a median symptom-onset age of 53 years (range, 13–75 years). The clinical presentations were either subacute (25%) or insidious (75%). The neurological symptoms varied from amnesia (80%), to brainstem disorder (75%), sleep disturbances (45%), delirium (40%), CA (35%), dysphagia (30%), psychosis (20%), depression (20%), seizures (10%), dysarthria (20%), and respiratory failure. The clinical signs of myoclonus (40%), exaggerated startle (30%), diffuse rigidity (30%), and hyperreflexia (30%) were ascribed to central hyperexcitability. Some patients also showed autonomic dysfunctions related to the gastrointestinal tract (45%), urinary bladder (35%), cardiac conduction system (15%), and thermoregulation (5%). Paraneoplasia (lymphoma) was detected in only 10% of the cohort. Immunotherapies were effective and culminated in substantial long-term recovery (18–68 months from symptoms onset). They included the combinations of IVIg, intravenous methyl prednisolone (IVMP), oral prednisone, plasma exchange, cyclophosphamide, and rituximab. Importantly, since CA with myoclonus can be the sole manifestation in this entity, patients with the combination of these manifestations should undergo measurement of anti-DPPX Ab [9].

The functions of Kv4.2 channels are dependent on two types of auxiliary subunits: the intracellular Kv4.2-channel-interacting proteins, and the extracellular DPPX [30]. The clinical features resemble those found in DPPX knock-out mice [31]. However, there are no patient-based physiological data that suggest any pathogenic actions of anti-DPPX Ab.

Table 2 summarizes the clinical features patients with anti-VGCC, anti-Caspr2, and anti-DPPX ataxias.

## 3. CA Associated with Autoantibodies against Synaptic Adhesion/Organizing Proteins

### 3.1. Anti-LGI1 Ataxia

Leucine-rich glioma-inactivated 1 (LGI1) is a secreted neuronal protein known to form a trans-synaptic complex comprising the presynaptic disintegrin and metalloproteinase domain-containing protein 23 (ADAM23), which interacts with VGKC Kv1, and postsynaptic ADAM22, which interacts with AMPA receptors [32]. The adult mouse shows high expression of LGI1 in the area of the neocortex and the hippocampus, but moderate expression in the cerebellum [33]. LGI1 is a major antigen in autoimmune limbic encephalitis, but rarely in IMCAs.

Irani and coworkers (2010) studied 55 patients with Abs to LGI1 (two-thirds were males, average age, 60 years) [34]. They presented mainly with cognitive impairments such as amnesia, confusion/disorientation, seizures, mood disorders, and sleep disorders with a subacute time-course. None of the patients had malignant tumors. The majority showed significant improvement in the modified Rankin scores following immunotherapy with IVIg, intravenous/oral corticosteroids or their combinations.

Although another subsequent study identified motor disorders including CA in 7 of 34 patients [35], it did not include detailed clinical information. One case report described one young adult with predominant gait ataxia, associated with disinhibited behaviors and visual hallucinations [36].

Due to the nature of the trans-synaptic complex, autoantibodies against LGI1 can disrupt presynaptic and postsynaptic functions, as demonstrated in both in vitro and in vivo studies using serum and CSF IgGs from the patients [32]. Interestingly, LGI1 IgGs reduced the number of Kv1.1 channels, resulting in an increase in presynaptic release of the transmitter. Simultaneously, LGI1 IgGs also decreased the number of AMAPA receptors, leading to dysregulated long-term potentiation and impairments of reversible memory in behaviors.

### 3.2. Anti-IgLON5 Ataxia

IgLON5 is an adhesion molecule widely distributed in the CNS [37]. One systematic study that examined the clinical manifestations of anti-IgLON5 disease [37], identified four phenotypes: (1) sleep disorder with parasomnia and sleep breathing difficulty in eight (36%) patients; (2) bulbar syndrome including dysphagia, sialorrhea, stridor, or acute respiratory insufficiency in six (27%); (3) syndrome resembling progressive supranuclear palsy (PSP-like) in give (23%); and (4) cognitive decline with or without chorea in three (14%). The condition was characterized by the association of (HLA)-DRB1 *10:01 and HLA-DQB1 *05:01 alleles [37]. Notably, postmortem examination showed a novel neuronal tauopathy predominantly involving the hypothalamus and brainstem tegmentum [38]. The postmortem examinations were in two patients and had similar findings, suggesting the convergence of neurodegenerative and autoimmune diseases. The prognosis was poor. No association with malignancy was documented. Only 10% of the patients demonstrated mild and transient improvement following immunotherapy with corticosteroids and IVIg.

The main clinical feature of anti-IgLON5 ataxia is gait instability. Although disequilibrium was documented as the reason for the instability [37], the exact mechanism remains unclear. The gait instability was attributed to cerebellar dysfunction in some patients [37,39,40].

### 3.3. Anti-GluR Delta Ataxia

Glutamate receptor delta (GluRδ) is a postsynaptic transmembrane protein, expressed mainly in Purkinje cells (PCs). The molecular complex of neurexin-cerebellin-GluRδ is a synaptic adhesion molecule found in parallel fiber (PF)-PC synapses [41].

The association of pure types of CA with anti-GluRδ Ab is found mainly in children (from 18 months to 13 years of age) [42,43]. While these patients were negative for paraneoplastic conditions, their CA was preceded by infection or history of vaccination. Gait ataxia was prominent, associated with dysarthria and limb ataxia. Mostly good outcomes were reported following IVMP immunotherapy.

One study demonstrated that injection of polyclonal Abs against the putative ligand-binding site of GluRδ2 induced endocytosis of AMPA receptors in cultured PC cells, while subarachnoid administration of the same Ab induced ataxic movement in mice [44].

The clinical features of CA associated with autoantibodies against synaptic adhesion/organizing proteins are summarized in Table 2.

## 4. CA Associated with Autoantibodies against Transmitter Receptors

### 4.1. Anti-NMDA R and Anti-AMPA R Ataxias

Anti-NMDA R encephalitis affects usually young adults and children, predominantly in females [45,46]. Most patients develop prodromal symptoms, such as headache, fever, or nausea, followed by multistage symptoms ranging from a state of psychosis, memory deficits, seizures, and language disintegration to a state of unresponsiveness with catatonic features often associated with abnormal movements, autonomic disorders, and breathing instability [45]. The association of ovarian tumors (usually teratoma) is age dependent. For example, 45% of females (12–45 years) had an underlying tumor, whereas only 6% of females younger than 12 years had a tumor [47]. Approximately 75% of the patients showed substantial recovery after first-line immunotherapy (IVIg, corticosteroids, or plasma exchange) [45,46,47]. Although anti-NMDA R Abs bind to granule cells in the cerebellar cortex, cerebellar complaints were identified in only 5% of the patients [48].

In in vitro studies of hippocampal slices, application of patients’ CSF or serum IgGs caused a titer-dependent, reversible decrease of synaptic NMDAR through the mechanism of crosslinking and internalization [45,46].

Anti-AMPA R Ab-associated encephalitis is well characterized, although the number of reported patients is far less than that of anti-NMDAR encephalitis [49]. Clinical manifestations are divergent, and thus are classified into four phenotypes: (1) distinctive limbic encephalitis (short-term memory loss, confusion, abnormal behavior, and seizures), (2) limbic dysfunction and multifocal encephalopathy (seizures, psychiatric manifestations, CA, abnormal movements), (3) limbic encephalopathy preceded by motor deficits, such as weakness, and (4) psychosis with bipolar features [50]. One study found malignant tumors (lungs, breasts, and thymoma) in about 70% of the cohort [47]. The response to immunotherapy and anti-tumor therapy varied from full or partial improvement to no response. Although AMPA R is widely distributed in the cerebellum (for example, more than 10^5^ PF synapses on a PC), CA was found in only in 14% of the cohort.

One in vitro study demonstrated that application of patients’ CSF to laboratory animal hippocampal slices elicited a decrease in the number of GluR2-containing AMPAR clusters at the synapse [51].

### 4.2. Anti-mGluR1 Ataxia

The association of anti-mGluR1 Ab was described in two patients with PCDs (malignant lymphoma) [52]. However, subsequent studies identified anti-mGluR Ab in non-paraneoplastic ataxic patients [53]. A large-scale study examined the clinical course in 30 patients [54]. The main neurological manifestations were subacute cerebellar gait and limb ataxias in 86% of the patients, sometimes associated with extra-cerebellar symptoms, including behavioral changes (irritability, apathy, mood, personality change, psychosis with hallucinations, and catatonia), cognitive changes (memory problems, executive functions, and spatial orientation deficits) or dysgeusia. Seizures were not commonly found. About 10% of the patients had paraneoplastic conditions, such as cutaneous T lymphoma and Hodgkin’s lymphoma. The patients received immunotherapies, including IVIg, steroids, mycophenolate mofetil, cyclophosphamide, and rituximab, alone or in combination. A total of 40% of the patients showed significant improvement or complete resolution of symptoms, while 52% showed stabilization or mild improvement.

One study showed that application of patients’ serum IgGs blocked glutamate-stimulated formation of inositol phosphates in mGluR1α-expressing Chinese hamster ovary cells and induced ataxic movement in mice [52].

### 4.3. Anti-mGluR2 Ataxia

Two patients with anti-mGluR2 Ab and CA were reported recently [55]. Both had tumors (small cell tumor of unknown origin and alveolar rhabdomyosarcoma). The former patient died, while the latter patient showed good therapeutic response to IVMP and IVIg. Interestingly, CSF from these patients did not change the density of mGluR2 in cultures of rat hippocampal neurons.

### 4.4. Anti-mGluR5 Ataxia

Patients with anti-mGluR5 encephalitis exhibit a wide range of clinical features. One systematic study of 11 patients reported psychiatric (91%) and cognitive (91%) deficits, movement disorders (64%), sleep dysfunction (64%), and seizures (55%) [56]. In these patients, ataxia was described in only two patients (gait instability in one patient and ataxia in another). However, paraneoplastic conditions (mainly, Hodgkin lymphoma) were detected in 55% of the cohort. The response to immunotherapy was described as good. Application of serum IgGs obtained from these patients decreased the density of mGluR5 in cultures of rat hippocampal neurons.

### 4.5. Anti-GABA_A_ R and anti-GABA_B_ R Ataxias

Anti-GABA_A_ R Ab- and anti-GABA_B_ R Ab-associated encephalitides are well described [57,58,59,60]. However, CA is rarely observed in patients with anti-GABA_A_ R and anti-GABA_B_ R-associated encephalitis, and the pure type of CA with these autoantibodies is also rare.

The clinical features of anti-GABA_A_ R Ab-associated encephalitis include seizures, memory and cognitive deficits, behavioral changes, and psychosis, with variable time-course from acute to chronic [57]. Paraneoplastic conditions are infrequent [47]. Most of patients showed full or partial recoveries with immunotherapies. A few case reports have described the association of CA with the clinical features of limbic encephalitis. One case report described a 3-year-old boy with wide-ranging neurological features, including acute-onset confusion, lethargy, dystonic tongue movements, chorea, opsoclonus, and CA, followed by complex partial seizures and status epilepticus [57]. Past history was negative for infection or paraneoplasia. Another reported case was a 15-month-old boy who showed irritability, focal motor refractory seizures, choreoathetosis, CA, and dysphagia after herpes simplex virus type 1 (HSV1) encephalitis [58].

The clinical features of anti-GABA_B_R Ab-associated encephalitis include seizures, confusion and memory loss, associated with behavioral changes or psychosis with a subacute time-course [59], but is not usually associated with CA. One study showed that half of the 15 patients had paraneoplastic conditions (lung tumors) [59]. The response to immunotherapy and anti-tumor therapy varied from full or partial improvement to no response. Although considered a rare case, one report described a 64-year-old man who received adjuvant therapy with interferon-alpha for malignant melanoma who later developed CA without seizures [60].

The clinical features suggest that both types of Abs inhibit GABA_A_ R and GABA_B_ R. One experimental study reported that CSF obtained from patients with anti-GABA_A_ R Ab-associated limbic encephalitis had no effects on the total density of GABA_A_ R, including synaptic and extrasynaptic receptors [57]. This is in contrast to the effects of other Abs, such as anti-NMDA R Ab and anti-AMPA R Ab, which induced a decrease in the receptors in both synaptic and extrasynaptic sites, suggesting a functional blockade of GABA_A_ R [57].

### 4.6. Anti-Glycine R Ataxia

Glycine receptors are distributed mainly in the spinal cord, brainstem, and cerebellum. Autoantibodies toward GlyR were first reported in 2008 in a single patient with progressive encephalomyelitis, rigidity, and myoclonus (PERM) [61]. The clinical features of PERM in this patient included Stiff-Person syndrome (SPS, characterized by stiffness of the axial and lower limb muscles), brainstem signs, hyperekplexia (brainstem myoclonus or excessive startle), and other neurological defects [61]. CA is one of these diverse clinical features. In a systematic study of 45 patients with anti-Glycine R Ab, limb and gait ataxias were identified in 13% of the patients [62]. Furthermore, 20% of the patients had paraneoplastic conditions. Patients with anti-Glycine R Ab-associated disease generally show good response to immunotherapy, including different combinations of IVMP, oral prednisone, IVIg, and plasma exchange.

One study that used patients’ serum IgGs, including anti-Glycine R Ab, showed reduced Glycine R clusters in HEK cells, suggesting the pathogenic action of internalization [62].

The clinical features of anti-transmitter receptors ataxias are summarized in Table 2.

## 5. CA Associated with Autoantibodies against Myelin-Related Proteins

### Anti-MAG Ataxia

Myelin-associated glycoprotein (MAG) is a bifunctional molecule known to promote outgrowth as well as inhibit regeneration of both Schwann cells and the CNS [63]. The association of anti-MAG Ab with distal demyelinating neuropathy is well established in patients with monoclonal gammopathy of unknown significance [64].

CA associated with anti-MAG Ab was only described recently in five patients [64]. All patients had IgM gammopathy, and four of the five showed clinically evident neuropathy. MR spectroscopy showed cerebellar involvement, which diminished after improvement of CA with rituximab [64].

## 6. CA Associated with Autoimmunity against Glial Cells and Perivascular T Cells in the Brainstem

### 6.1. Autoimmune GFAP Astrocytopathy

Glial fibrillary acidic protein (GFAP) is the main intermediate filament in mature astrocytes and a component of their cytoskeleton [14]. Autoimmune GFAP astrocytopathy is characterized by fever, headache, convulsions, delirium, meningism, loss of visual acuity, and ataxia [65,66]. The median age at onset is 40–50 years [65,66]. Based on these features, this condition is often referred to as encephalitis, meningoencephalitis, myelitis, or their combinations [14]. CA was described as accompanying meningoenchephalomyelitis (40%) [14]. Atypical patients with progressive CA, proximal myoclonus, and bulbar symptomatology have also been reported [14].

Autoimmune GFAP astrocytopathy is often associated with other autoimmune diseases, such as type 1 diabetes, thyroiditis, or even other types of autoimmune encephalitis (NMDA encephalitis) [65,66]. Paraneoplastic conditions were identified in a third of the patients [67], and history of upper respiratory tract infection was reported in 40% of the patients [66]. CSF studies showed inflammatory changes (monocytic pleocytosis), high protein and low glucose levels [65,66]. Brain and spinal cord MRI studies showed nonspecific findings, such as T2-hyperintense lesions, gadolinium-enhanced lesions in T1 sequences, and even leptomeningeal enhancement in half of the patients but no changes in the other half of the patients [66].

IVMP is usually effective in acute treatment [66], although some patients also require additional treatment, such as plasma exchange or IVIg. Maintenance therapy using mycophenolate mofetil, azathioprine, or rituximab is necessary in 20–50% of the patients in order to prevent relapse [66].

### 6.2. CLIPPERS

Chronic lymphocytic inflammation with pontine perivascular enhancement responsive to steroids (CLIPPERS) is characterized by marked perivascular T-cell inflammation [15,68]. The lymphocytes, mainly CD4-dominant lymphocytes, infiltrate the perivascular area. The inflammatory infiltration involves basically the white matter, but also the grey matter, of the pons and adjacent rhombencephalic structures, such as the cerebellar peduncles, cerebellum, medulla, and the midbrain [15,68]. The perivascular infiltration appears as a characteristic pattern of gadolinium enhancement on MRI, with multiple “punctate” and/or “curvilinear” gadolinium-enhancing lesions, resulting in “peppering” of the pons, with or without peripontine lesions [15,68]. Since characteristic Abs have not been reported, the characteristic MRI findings are only clues to a diagnosis. The condition affects mainly men in their 50s [15,68]. The affected patients show subacute onset of the combination of brainstem symptoms and CA, which include pancerebellar ataxia, dysarthria, dysphagia, dysgeusia, oculomotor abnormalities, altered facial sensation, facial nerve palsy, vertigo, pyramidal signs, and sensory disorders [15,68]. CLIPPERS responds well to corticosteroids [15,68]. Intravenous methyl prednisolone is followed by maintenance immunotherapy with the combination of oral prednisolone and corticosteroid-sparing immunosuppressants [15,68]. Since the clinical course is relapsing–remitting in nature, long-term maintenance therapy is necessary [68].

## 7. Primary Autoimmune Cerebellar Ataxia (PACA)

When immune-mediated mechanisms are strongly suspected but serological tests do not match any of the established etiologies, the condition is classified under a spectrum of PACA [16,17]. In other words, the PACA spectrum serves as an umbrella that covers heterogeneous etiologies [16,17]. Despite its heterogeneous nature, common clinical features are found in patients with PACA [16,17]. They predominantly show subacute or acute course of pure cerebellar syndrome (mainly gait ataxia, associated with a variable degree of limb incoordination, dysarthria, and nystagmus). The MRI at presentation is usually normal or may show primarily vermian atrophy. MR spectroscopy of the cerebellum shows clear involvement of the vermis, with or without involvement of the hemispheres.

Patients with PACA characteristically have features suggestive of autoimmune etiologies. First, HLA type DQ2 is significantly over-represented in these patients. This HLA type is over-represented in patients with certain autoimmune diseases (celiac disease, GA, type 1 diabetes mellitus, Stiff-Person syndrome, autoimmune thyroid disease, and autoimmune polyendocrine syndromes), given that this HLA marker is found in up to 35% of healthy subjects [16,17]. Second, history of other autoimmune disorders, or family history of autoimmune disorders is often present. Third, CSF examination shows evidence of autoimmune inflammation, including CSF pleocytosis, and/or positive CSF-restricted IgG oligoclonal bands. More directly, the majority of patients with PACA show an association with a variety of cerebellar neuronal Abs that are poorly characterized. Consistently, an immunohistochemical study show that cerebellar Abs were present in 60% of the patients with idiopathic sporadic ataxia [16]. Four different staining patterns were observed (cytoplasmic alone, cytoplasmic with processes, nuclear in Purkinje cells, and granular cells) [16]. Pathogenic Abs against ion channels/their related proteins, synaptic adhesion/organizing proteins, and transmitter receptors should not come under the category of PACA. Table 3 lists the autoantibodies associated with PACA [69,70,71,72,73,74,75,76,77,78,79,80,81,82,83,84,85,86]. However, it should be acknowledged that PACA includes seronegative CA with autoimmune nature.

Many types of immunotherapies have been used in PACA. One study showed that mycophenolate mofetil improved CA with parallel improvement in MR spectroscopy [87].

### 7.1. Molecular Characterization of Autoantigens Found in PACA

Autoantibodies target a great diversity of neuronal antigens, mostly located at intracellular levels. These autoantigens are involved in critical cellular functions, including Ca^2+^ signal pathways, clathrin-dependent endocytosis of AMPA receptors, exocytosis, such as synaptic vesicle release, cytoskeleton, and transcription factors (Table 3). The only clinical manifestation associated with anti-Homer 3, anti-CARP VIII, anti-PKC-γ, and anti-TRIM9/67 autoantibodies is CA. Notably, Homer 3, CARP VIII, and PKC-γ appear to play essential roles in molecular pathways linked to cerebellar specific synaptic plasticity. In contrast, other autoantibodies show a wider range of clinical phenotypes. These autoantigens are involved in various cell functions throughout the CNS.

### 7.2. Significance of Non-Specific Autoantibodies Commonly Found in Autoimmune Diseases

Anti-thyroid, anti-SS_A_/SS_B_, and low-titer anti-GAD Abs are sometimes found in PACA. However, clinicians should evaluate the significance of these non-specific autoantibodies, which can also be associated with other autoimmune diseases [88]. Furthermore, the same autoantibodies are also present in healthy subjects [88]. Thus, these autoantibodies may co-exist accidentally and may not reflect the autoimmunity that insults the cerebellum [88].

Hashimoto’s encephalopathy (HE) was originally classified as an independent entity [89]. It was reported to be associated with anti-thyroid Abs and to respond satisfactorily to steroids [89]. Autoantibodies against the NH_2_-terminal of α-enolase (anti-NAE Ab) have also been described [90]. The recently reported association with other autoantibodies (e.g., low titer of anti-GAD Ab) [3], makes a single entity unlikely. Thus, these patients could be described as patients with a good response to corticosteroids and classified as one phenotype of PACA [3].

Taken together, the above non-specific autoantibodies commonly found in autoimmune diseases do not seem to point to a specific etiology, but rather to an autoimmune diathesis.

## 8. Conclusions

Based on the above review, we highlight two important aspects in the diagnosis of IMCAs. The Figure shows these two factors: “CA as predominant clinical feature or part of a more diverse presentation” and “Well-characterized or poorly characterized autoantibodies” (Figure 1). Given the potential response to therapies [91], search for rare forms of immune ataxias should be performed once the common causes have been excluded.

### 8.1. CAs as Predominant Clinical Feature or Part of a More Diverse Presentation

In most cases, the rare entities described here refer to a more global neurological dysfunction where CA can be one of many neurological features. It is useful to consider whether CA is the primary phenotype (especially, isolated) and whether the frequency of CA is high in each subtype. This may determine the clinical approach in serological testing for these entities.

PACA is characterized by the predominant development of acute or subacute pure cerebellar syndrome and as such future identification of novel antibodies for patients labeled as having PACA are likely to be cerebellar specific. In contrast, since ion channels and related proteins, synaptic adhesion/organizing proteins, transmitter receptors, and glial cells are distributed throughout the central and peripheral nervous systems, autoimmunity toward these nonspecific molecules may explain the development of diverse neurological features.

### 8.2. Well-Characterized or Poorly Characterized Autoantibodies

Autoantibodies are helpful in the diagnosis of IMCA. However, the significance of specific autoantibodies should be examined comprehensively. The presence of well-characterized Abs suggests autoimmunity-related cerebellar insult. In contrast, the presence of poorly characterized Abs, sometimes found in PACA, may reflect co-existence of autoimmunity, and such autoimmunity defined by the antibody (e.g., thyroid antibodies) may not necessarily be the cause of the cerebellar insult. For a comprehensive assessment, the diagnostic algorithm proposed for PACA will be helpful [17].

## Figures and Tables

**Figure 1 brainsci-12-01165-f001:**
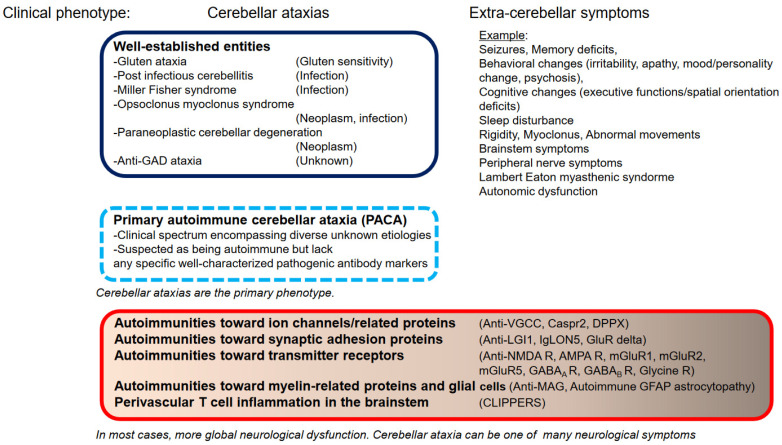
A schematic diagram of the diverse etiologies in immune-mediated cerebellar ataxias. In case of well-establish etiologies, cerebellar ataxia is the primary presentation. Primary autoimmune cerebellar ataxia (PACA) encompasses etiologies associated with poorly characterized nonpathogenic antibodies. By contrast, cerebellar ataxia can be one of many neurological presentations in autoimmunity towards ion channels and related proteins, synaptic adhesion molecules, transmitter receptors, and glial cells, or in T-cell inflammation in the brainstem. In some cases, ataxia may be rare.

**Table 1 brainsci-12-01165-t001:** Well-characterized or partly characterized autoantibodies in immune-mediated cerebellar ataxias.

Associated Autoantibody	Suggested Specific Etiology in Immune-Mediated Cerebellar Ataxias
Anti-gliadin, TG 2, 6	Gluten ataxia
Anti-GQ1b	Miller Fisher syndrome
Anti-PCA-1/Yo	PCDs; Breast, uterus and ovarian carcinomas
Anti-ANNA-1/Hu	PCDs; Small cell lung carcinoma
Anti-CV2/CRMP5	PCDs; Small cell lung carcinoma, thymoma
Anti-ANNA2/Ri	PCDs and Paraneoplastic OMS; Breast carcinoma
Anti-Tr/DNER	PCDs; Hodgkin’s lymphoma
Anti-Ma2	PCDs; Testis and lung carcinoma
Anti-AGNA/SOX1	PCDs; Small cell lung carcinoma
Anti-ZIC4	PCDs; Small cell lung carcinoma
Anti-amphiphysin	PCDs; Small cell lung and breast carcinomas
Anti-PCA-2/MAP1B	PCDs; Small cell lung carcinoma, non-small cell lung carcinoma
Anti-ANNA-3/DACH1	PCDs; Small cell lung carcinoma
Anti-KLHL11	PCDs; testicular carcinoma

In PCD, this table shows the antibodies classified by Sergio and Jerome as “major antibodies in PCD” [6]. In PCDs, these antibodies are shown in order of frequency according to the study of Honnorat [2]. PCDs: paraneoplastic cerebellar degeneration, DNER: delta/notch-like epidermal growth factor-related receptor, SOX1: sex determining region Y-related high-mobility group box 1, MAP1B: microtubule associated protein 1B, KLHL11: Kelch-like protein 11.

**Table 2 brainsci-12-01165-t002:** Clinical features of cerebellar ataxia associated with autoantibodies against ion channels and related proteins, synaptic adhesion/organizing molecules, and transmitter receptors.

Associated Autoantibodies	Main Phenotypes	Cerebellar Ataxias	Trigger of Autoimmunity	Therapeutic Outcome
** *Autoantibodies against ion channels and related proteins* **				
Anti-VGCC	CAs or/and Lambert-Eaton myasthenic syndrome	Main phenotype	Paraneoplastic (SCLC). Sometimes non-paraneoplastic	Mostly good of not paraneoplastic
Anti-Caspr2	Multi focal symptoms: cognitive disturbance (79%) and epilepsy (53%), CA (35%), insomnia (68%), neuropathic pain (61%), peripheral nerve hyperexcitability (54%), autonomic dysfunction (44%), and weight loss (58%)	Sometimes as one of multitude of features (35%). Pure CA or episodic ataxia	Non-paraneoplastic. Infrequent paraneoplastic (20%, mostly thymoma)	Variable. Good recovery in 50% of patients
Anti-DPPX	Multi focal symptoms. Amnesia (80%), brainstem disorder (75%), sleep disturbances (45%), delirium (40%), myoclonus (40%), CA (35%)	Sometimes as one of multitude of features (35%). Few, pure CA	Non-paraneoplastic. Infrequent paraneoplastic (10%, lymphoma)	Mostly good
** *Autoantibodies against synaptic adhesion/organizing molecules* **				
Anti-LGI1	Limbic encephalitis. Amnesia, confusion/disorientation, seizures	Rare, as one of multitude of features	Mostly non-paraneoplastic	Mostly good
Anti-IgLON5	Sleep disorders (36%), bulbar syndrome (27%), syndrome resembling progressive supranuclear palsy (23%), and cognitive decline (14%)	Gait instability might be attributed to ataxia in some patients?	Mostly non-paraneoplastic. Association of taupathy in the hypothalamus and brainstem tegmentum	Poor
Anti-GluR delta	CA	Main phenotype	Infection	Good
** *Autoantibodies against transmitter receptors* **				
Anti-NMDA R	Multi focal symptoms. Psychosis, memory deficits, seizures, and language disintegration, followed by a state of unresponsiveness	Rare, as one of multitude of features	Non-paraneoplastic. Paraneoplastic, depending on age, 10–45%	Mostly good. Good recovery in 75% of patients
Anti-AMPA R	Multi focal symptoms. Distinctive limbic encephalitis (short-term memory loss, confusion, abnormal behavior, and seizures), limbic dysfunction and multifocal encephalopathy (seizures, psychiatric manifestations, CA, abnormal movements), limbic encephalopathy preceded by motor deficits, and psychosis	Infrequent, as one of multitude of features (14%)	Paraneoplastic (70%, lung, breast, thymoma). Sometimes non-paraneoplastic	Variable
Anti-mGluR1	CA (86%) associated with behavioral changes (irritability, apathy, mood, personality change, psychosis with hallucinations, and catatonia), cognitive changes (memory problems, executive functions and spatial orientation deficits) or dysgeusia	Main phenotype	Non-paraneoplastic. Infrequent paraneoplastic (10%, lymphoma)	Variable. Good recovery in 50% of patients
Anti-mGluR2	CAs	Main phenotype	Paraneoplastic	Variable
Anti-mGluR5	Multi focal symptoms. Psychiatric (91%), cognitive (91%), movement disorders (64%), sleep dysfunction (64%), and seizures (55%)	Infrequent, as one of multitude of features (18%)	About the same. Paraneoplastic and Non-paraneoplastic	Mostly good
Anti-GABA_A_ R	Multi focal symptoms. seizures, memory and cognitive deficits, behavioral changes, and psychosis	Rare, as one of multitude of features	Mostly non-paraneoplastic	Good
Anti-GABA_B_ R	Multi focal symptoms. Behavioral changes or psychosis	Rare, as one of multitude of features	About the same. Paraneoplastic and Non-paraneoplastic	Variable
Anti-Glycine R	Progressive encephalomyelitis with rigidity and myoclonus, sometimes associated with brainstem symptoms, brainstem myoclonus, excessive startle, and CA	Infrequent, as one of multitude of features (13%)	Non-paraneoplastic. Infrequent, paraneoplastic (20%)	Mostly good

VGCC: voltage-gated calcium channel, Caspr2: Contactin-associated protein-like 2, DPPX: Dipeptidyl-peptidase-like protein-6, GluR delta: glutamate receptor delta, R: receptor, mGluR: metabotropic glutamate receptor, CA: cerebellar ataxia, SCLC: small cell lung carcinoma.

**Table 3 brainsci-12-01165-t003:** Clinical features of primary autoimmune cerebellar ataxia associated with no characterized autoantibodies.

Targeted Antigens	Nature of Autoantigens	Frequency of Cerebellar Ataxia	Extracerebellar Symptoms	Association with Neoplasia	References
Sj/ITPR-1	ITPR-1 triggers Ca^2+^ release from smooth ER, as the main intracellular Ca^2+^ store in response to stimulation of mGluR1.	Sometimes, as one of multitude of clinical features (8/22)	Peripheral neuropathy, encephalitis, myelopathy	Sometimes (breast and others carcinomas)	[69,70]
Homer-3	Homer-3 cross-links cytoplasmic C-terminus of mGluR1.	Main phenotype (2/2)	-	No reports	[71,72]
CARP VIII	CARP VIII reduces ITPR1 affinity for IP3 by binding to the modulatory domain (residues 1387 to 1647) of that receptor	Main phenotype (3/3)	-	Mostly (melanoma, ovary carcinoma)	[73,74,75]
PKC-γ	Upon binding of cytosolic Ca^2+^ released by ITPR1, PKC-γtranslocates to the plasma membrane to modulate the function of other proteins.	Main phenotype (2/2)	-	Mostly (non-small cell lung carcinoma)	[76,77]
Neurochondrin	Neurochondrin negatively regulates phosphorylation of Ca^2+^/calmodulin-dependent protein kinase II (CaMKII) and functions as a mediator of neurite growth and synaptic plasticity	Main phenotype (3/3)	Dystonia	No reports	[78,79]
Ca/ARGHAP26	ARGHAP26 is involved in clathrin-dependent endocytosis.	Main phenotype (7/10)	Cognitive impairment, hyperekplexia	Sometimes (diverse carcinomas, B-cell lymphoma, melanoma)	[80]
Septin-5	Septin-5 regulates exocytosis.	Main phenotype (4/4)	Prominent eye movement symptoms (oscillopsia or vertigo).	No reports	[81]
Nb/AP3B2	AP3B2 regulates the levels of selected membrane proteins in synaptic vesicles.	Sometimes, as one of multitude of clinical features (4/9)	Sensory ataxia, paresthesia, weakness	No reports	[82,83]
TRIM9/67	TRIM9 interacts with membrane-associated SNARE protein, SNAP25, and acts as negative regulator of exocytosis during axonal branching and growth.	Main phenotype (2/2)	-	Mostly (non-small cell lung carcinoma)	[84]
TRIM46	TRIM46 localizes to the axon initial segment and plays an instructive role in the initial polarization of neuronal cells.	Sometimes, main phenotype (1/3)	Encephalomyelitis, dementia	Mostly, (small cell lung carcinoma)	[85]
Neuronal intermediate filament (light chain)	Neuronal intermediate filament is cytoskeletal structural components in large-diameter myelinated axons.	50% (11/21)	Encephalopathy, myelopathy	Mostly (neuroendocrine carcinomas)	[86]

SOX1: sex determining region Y-related high-mobility group box 1, CARP VIII: Carbonic anhydrase-related protein VIII, PKCγ: Protein kinase C gamma, Ca/ARHGAP26: Ca/Rho GTPase-activating protein 26, Nb/AP3B2: Nb/adaptor complex 3 B2, TRIM: axon initial segment protein tripartite motif.

## Data Availability

The concept reported in this manuscript is not associated with raw data. No software application or custom code was used to generate the information reviewed in this paper.

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
