# Peer review of "Rare Etiologies in Immune-Mediated Cerebellar Ataxias: Diagnostic Challenges"

_brainsci, 2022, doi:10.3390/brainsci12091165_

Round 1
Reviewer 1 Report
This manuscript is a review of less common etiologies of immune-mediated cerebellar ataxias and in general does a thorough job of covering this area. The tables provide a good summary of what is covered in the article and a quick reference source.
The only issue to be addressed is clarity of expression in a number of instances. Some are grammatical and syntactical and others are just awkward or incomplete expressions of the concepts the authors wish to convey. Since there is no mechanism in the review process for uploading an annotated manuscript, I will list the areas in question by the line of the manuscript.
Line 19: there is a list of relatively specific antigenic targets that ends with “brainstem”. What is unique to brainstem as opposed to other CNS structures that makes it a specific antigenic target? Could you express it as “uncharacterized antigens in the brainstem”?
Lines 20-22: ”The clinical entity of primary autoimmune cerebellar ataxia (PACA) describes various conditions with autoimmune-suspected predominant CAs even in the lack of specific well-characterized pathogenic antibody markers.” This sentence is confusing and needs to be clarified. Suggestion: “The term primary autoimmune cerebellar ataxia (PACA) refers to ataxic conditions suspected to be autoimmune even in the absence of specific well-characterized pathogenic antibody markers.”
Lines 78-79: “The aim of this review is to clarify the position of recently reported subtypes and spectrum of PACA within the IMCAs.” This could be improved by changing to “positions of recently reported subtypes within the IMCAs as well as the spectrum of PACA.”
Lines 92-94: “Another study of 16 anti-VGCC Ab-positive patients with PCDs and SCLC found no differences in the clinical profiles of PCD patients with or without anti-VGCC Ab.” If the study was of anti-VGCC Ab-positive patients, how were the patients without antibodies obtained? I assume by historical controls, but this should be clarified.
Line 94: “Of these patients, one showed complete recovery” I assume this was response to therapy, but “recovery” implies that it might be spontaneous. This should be clarified.
Line 113: “Kv.1 proper positioning” “Proper positioning of Kv.1” is a better way to phrase this.
Line 116: “ In a study of 38 cohort with anti-Caspr2 Ab-associated encephalitis” “In a study of 38 patients with anti-Caspr2 Ab-associated encephalitis---”
Line 126: Stereotypic instead of stereotype. CA instead of CAs. There is frequent use of the abbreviation CAs throughout the manuscript, where CA would be more appropriate. A patient does not suffer from ataxias, but rather from ataxia. So unless the reference is definitely to a spectrum of ataxic conditions CA is the better term.
Line 137-138: “Permanent CAs were observed in 73% of the LE/+ patients, whereas episodic CAs were found in 55% of these patients.” This statement is confusing. If 73% had permanent ataxia, how could more than half of the patients also have episodic ataxia? Did episodic ataxia progress to permanent ataxia in some cases?
Lines 138-139: “These patients showed Human Lymphocyte Antigen (HLA) -DRB1*11:01”. To which patients are the authors referring, all patients with the antibodies or only the ones who developed ataxia, and was that HLA antigen seen in all of “these” patients?
Line 145: “Another in vitro study reported that anti-Caspr2 Ab from these patients” Was there a previous in vitro study? Why is this “another in vitro study”. Again, who are “these” patients.
Line 148: “clustering of juxtaparanodes and hyperexcitability” Change to clustering of juxtaparanodes and ”in” hyperexcitability, since hyperexcitability is not being clustered.
Line 151: “Kv4.2, is the target antigen in autoimmune limbic encephalitis.” Change to “a target antigen” unless it is the only antigen associated with autoimmune limbic encephalitis.
Line 167: typo “from” not “form”
Table 2: Anti-mGluR5 the incidence of ataxia should not be listed as “rare” if it is 18%, especially since several other entities with lower percentages (13% and 14%) are listed as “infrequent”
Line 189: “ in the area of the neocortex, especially the hippocampus” Change to “in the neocortex and the hippocampus
Line 190: Is this antigen the major Ag or “an” Ag causing limbic encephalitis?
Line 192: “55 patients with LGI1” Change to “patients with antibodies to LGII”.
Lines 193-194: “They presented mainly with amnesia, confusion/disorientation, and seizures, with a subacute time-course, sometimes associated with sleep disorders” The sleep disorders are listed separately from the other symptoms, did they have a different time course? If not, the time course should be mentioned after the entire list of symptoms.
Lines 202-203: “autoantibodies-induced dysfunction of LGI1 can disrupt presynaptic and postsynaptic functions,” Change to “autoantibodies against LGI1 can disrupt presynaptic and postsynaptic functions”.
Line 205: “LGI1 IgGs reduced the number of Kv1.1, resulting in an increase in presynaptic release”
Either say “reduced the amount of Kv1.1” or “reduced the number of Kv1.1 channels”. The presynaptic release of what?
Line 216: The postmortem examinations were in two patients and had similar findings. It would be good to point that out.
Line 230: “pure type of CAs” Use “pure types of CA (or CAs)” .
Lines 247-248: “The association of ovarian tumors (usually teratoma) is age-dependent (10-45%)” This sentence does not make sense. First of all, what is the age-dependent difference (old vs. young) and secondly, how can age-dependence be expressed as a range of percentages?
Lines 256-261: It would be helpful to label the four phenotypes with numbers or letters or at least use semicolons to separate them.
Line 279: Clarify who “they” are. All patients or the ones with lymphoma?
Line 291: “Interestingly, CSF from these patients did not change the density of mGluR2.” In what system was this studied?
Line 299: Same question. In what system?
Lines 301-303: two issues of verb-subject agreement. It can be either “encephalitis is” or “encephalitides are”. Likewise, “pure types of CAs” or “pure type of CA”.
Line 308: “Most of patients showed full or partial recoveries to immunotherapies” Most of these patients showed full or partial recovery with immunotherapy.
Line 318: “not usually associated with CAs.” Change to “but is not usually associated with CA”
Line 318: “Half the patients” Half of how many patients? Are these data from reference 59? It would be good to give an idea of how many patients were in the study.
Line 357: typo, this should be an italicized subtitle
Line 363: “CAs were described as accompaniment with meningoenchephalomyelitis (40%)” Change to “CA was described as accompanying meningoencephalomyelitis”.
Lines 390-391: Individual patients have ataxia not ataxias.
Line 407: “have” is a better term than “develop”, since they don’t develop specific HLA types.
Lines 409-411: “This HLA type is observed in certain autoimmune diseases (celiac disease, GA, type 1 diabetes mellitus, Stiff-Person syndrome, autoimmune thyroid disease, and autoimmune polyendocrine syndromes), but not in healthy subjects”. This statement would imply that the DQ2 HLA type is never seen in anyone without autoimmune disease.
Lines 416-417: “Consistently, immunohistochemical studies show various staining patterns (60% of the patients)” Staining patterns for what in which system with which antibodies? What does 60% mean in this context, particularly when coupled with the term consistently in the same sentence?
Lines 420-421: “However, it should be acknowledged that PACA include seronegative CAs with auto-immune nature”. PACA includes seronegative CAs with auto-immune etiologies.
Lines 429-431: “Anti-Homer 3, anti-CARP VIII, anti-PKC-γ, and anti-TRIM9/67 Abs are related exclusively to the clinical presentation of CAs.” This is hard to understand at first glance. Perhaps a better way to phrase it would be: The only clinical manifestation associated with anti-Homer 3, anti-CARP VIII, anti-PKC-γ, and anti-TRIM9/67 autoantibodies is cerebellar ataxia.
Line 454: The Figure in the Conclusions section does not look right. Something was lost in how it was put into the manuscript.
Table 3: Under “nature of autoantigens”
PKC-gamma : remove “is”
Neuronal intermediate filament: change to “cytoskeletal structural component”
Header in column 2: Should be “Frequency of Cerebellar Ataxia”
Author Response
We thank the Reviewer for the careful reading and the positive comments. Please find below our reply. The edited portion is highlighted by yellow markers in the text.
Line 19: there is a list of relatively specific antigenic targets that ends with “brainstem”. What is unique to brainstem as opposed to other CNS structures that makes it a specific antigenic target? Could you express it as “uncharacterized antigens in the brainstem”?
Reply. We agree with the comment. Accordingly, we changed this sentence as follows:
“Apart from these well-established entities, cerebellar ataxia (CA) occurs also in association with autoimmunity against ion channels and related proteins, synaptic adhesion/organizing proteins, transmitter receptors, glial cells, as well as the brainstem antigens. Uncharacterized antigens of the brainstem can also be considered, but further studies are required by definition in order to establish these antigens as compared to those found in the cerebellar circuitry”
Lines 20-22: ”The clinical entity of primary autoimmune cerebellar ataxia (PACA) describes various conditions with autoimmune-suspected predominant CAs even in the lack of specific well-characterized pathogenic antibody markers.” This sentence is confusing and needs to be clarified. Suggestion: “The term primary autoimmune cerebellar ataxia (PACA) refers to ataxic conditions suspected to be autoimmune even in the absence of specific well-characterized pathogenic antibody markers.”
Reply. We agree with the criticism. Accordingly we changed this sentence as follows:
“The term primary autoimmune cerebellar ataxia (PACA) refers to ataxic conditions suspected to be autoimmune even in the absence of specific well-characterized pathogenic antibody markers.”
Lines 78-79: “The aim of this review is to clarify the position of recently reported subtypes and spectrum of PACA within the IMCAs.” This could be improved by changing to “positions of recently reported subtypes within the IMCAs as well as the spectrum of PACA.”
Reply. Based on the suggestion, we changed this sentence as follows:
“The aim of this review is to clarify positions of recently reported subtypes within the IMCAs as well as the spectrum of PACA.”
Lines 92-94: “Another study of 16 anti-VGCC Ab-positive patients with PCDs and SCLC found no differences in the clinical profiles of PCD patients with or without anti-VGCC Ab.” If the study was of anti-VGCC Ab-positive patients, how were the patients without antibodies obtained? I assume by historical controls, but this should be clarified.
Reply.
“Another study showed no difference in the clinical profiles between 16 anti-VGCC Ab-positive patients with PCDs and SCLC and nine anti-Hu Ab-positive patients with PCDs and SCLC.”
Line 94: “Of these patients, one showed complete recovery” I assume this was response to therapy, but “recovery” implies that it might be spontaneous. This should be clarified.
Reply. We agree the criticism. Accordingly, we changed this sentence as follows:
“Of these patients, one showed good responses to therapies”
Line 113: “Kv.1 proper positioning” “Proper positioning of Kv.1” is a better way to phrase this.
Reply. We appreciate suggestions. Accordingly, we changed this sentence as follows:
“Caspr2 forms a molecular complex with transient axonal glycoprotein-1 (TAG-1)/contactin-2, and VGKC Kv1 in compartments critical for neuronal activity and is required for proper positioning of Kv.1.”
Line 116: “ In a study of 38 cohort with anti-Caspr2 Ab-associated encephalitis” “In a study of 38 patients with anti-Caspr2 Ab-associated encephalitis---”
Reply. We agree with the comment. Accordingly, we changed this sentence as follows:
“In a study of 38 patients with anti-Caspr2 Ab-associated encephalitis…”
Line 126: Stereotypic instead of stereotype. CA instead of CAs. There is frequent use of the abbreviation CAs throughout the manuscript, where CA would be more appropriate. A patient does not suffer from ataxias, but rather from ataxia. So unless the reference is definitely to a spectrum of ataxic conditions CA is the better term.
Reply. We used a term of stereotypic instead stereotype. We agree that CA might be considered as a preferred term.
Line 137-138: “Permanent CAs were observed in 73% of the LE/+ patients, whereas episodic CAs were found in 55% of these patients.” This statement is confusing. If 73% had permanent ataxia, how could more than half of the patients also have episodic ataxia? Did episodic ataxia progress to permanent ataxia in some cases?
Reply. We agree with the comments. To avoid misunderstanding, we changed this sentence as follows:
“Permanent CAs were observed in 73% of the LE/+ patients. Episodic CAs were found in 55% of these patients who showed LE/+ (LE and extra-limbic symptoms).”
Lines 138-139: “These patients showed Human Lymphocyte Antigen (HLA) -DRB1*11:01”. To which patients are the authors referring, all patients with the antibodies or only the ones who developed ataxia, and was that HLA antigen seen in all of “these” patients?
Reply. We agree with the comments. To avoid misunderstanding, we changed this sentence as follows:
“These patients with LE showed Human Lymphocyte Antigen (HLA) -DRB1*11:01, had high serum titers of anti-Caspr2 Ab, and were positive for anti-Caspr2 Ab in CSF”
Line 145: “Another in vitro study reported that anti-Caspr2 Ab from these patients” Was there a previous in vitro study? Why is this “another in vitro study”. Again, who are “these” patients.
Reply. We agree the comment. Accordingly, we changed this sentence as follows:
“A study using a cell-based study reported that anti-Caspr2 Ab from the patients with LE and PNH did not induce internalization of Caspr2.”
Line 148: “clustering of juxtaparanodes and hyperexcitability” Change to clustering of juxtaparanodes and ”in” hyperexcitability, since hyperexcitability is not being clustered.
Reply. We agree the comment. Accordingly, we changed this sentence as follows:
“, which resulted in interference with the clustering of juxtaparanodes and in hyperexcitability”
Line 151: “Kv4.2, is the target antigen in autoimmune limbic encephalitis.” Change to “a target antigen” unless it is the only antigen associated with autoimmune limbic encephalitis.
Reply. We agree the comment. Accordingly, we changed this sentence as follows:
“Dipeptidyl-peptidase-like protein-6 (DPPX, DPP6), a regulatory subunit of VGKC Kv4.2, is a target antigen in autoimmune limbic encephalitis.”
Line 167: typo “from” not “form”
Reply. We appreciate the detailed check. The change has been made.
“(18–68 months from symptoms onset)”
Table 2: Anti-mGluR5 the incidence of ataxia should not be listed as “rare” if it is 18%, especially since several other entities with lower percentages (13% and 14%) are listed as “infrequent”
Reply. We thank the reviewer for the careful check. We changed from “rare” to “infrequent”.
Line 189: “ in the area of the neocortex, especially the hippocampus” Change to “in the neocortex and the hippocampus
Reply. We agree with the comment. Accordingly, we changed this sentence as follows.
“in the area of the neocortex and the hippocampus”
Line 190: Is this antigen the major Ag or “an” Ag causing limbic encephalitis?
Reply. Based on the comment, we changed this sentence as follows:
“LGI1 is a major antigen in autoimmune limbic encephalitis,”
Line 192: “55 patients with LGI1” Change to “patients with antibodies to LGII”.
Reply. Based on the comment, we changed this sentence as follows:
“55 patients with Abs to LGI1”
Lines 193-194: “They presented mainly with amnesia, confusion/disorientation, and seizures, with a subacute time-course, sometimes associated with sleep disorders” The sleep disorders are listed separately from the other symptoms, did they have a different time course? If not, the time course should be mentioned after the entire list of symptoms.
Reply. We agree with the comment. Accordingly, we listed symptoms in order of frequency as follows:
“They presented mainly with cognitive impairments such as amnesia, confusion/disorientation, seizures, mood disorders, and sleep disorders with a subacute time-course.”
Lines 202-203: “autoantibodies-induced dysfunction of LGI1 can disrupt presynaptic and postsynaptic functions,” Change to “autoantibodies against LGI1 can disrupt presynaptic and postsynaptic functions”.
Reply. We agree with the comment. Accordingly, we changed this sentence as follows:
“autoantibodies against LGI1 can disrupt presynaptic and postsynaptic functions”
Line 205: “LGI1 IgGs reduced the number of Kv1.1, resulting in an increase in presynaptic release”
Either say “reduced the amount of Kv1.1” or “reduced the number of Kv1.1 channels”. The presynaptic release of what?
Reply. Based on the comment, we changed this sentence as follows:
“Interestingly, LGI1 IgGs reduced the number of Kv1.1 channels, resulting in an increase in presynaptic release of the transmitter.”
Line 216: The postmortem examinations were in two patients and had similar findings. It would be good to point that out.
Reply. We agree with the comment. Accordingly, we added a following sentence:
“Notably, postmortem examination showed a novel neuronal tauopathy predominantly involving the hypothalamus and brainstem tegmentum [38]. The postmortem examinations were in two patients and had similar findings, suggesting the convergence of neurodegenerative and autoimmune diseases.”
Line 230: “pure type of CAs” Use “pure types of CA (or CAs)” .
Reply. Based on the comment, we changed this sentence as follows:
“The association of pure types of CA with anti-GluRδ Ab is found mainly in children”
Lines 247-248: “The association of ovarian tumors (usually teratoma) is age-dependent (10-45%)” This sentence does not make sense. First of all, what is the age-dependent difference (old vs. young) and secondly, how can age-dependence be expressed as a range of percentages?
Reply. We agree with the criticism. Accordingly, we added following sentences:
“The association of ovarian tumors (usually teratoma) is age-dependent. For example, 45 % of females (12-45 years) had an underlying tumor, whereas only 6% of females younger than 12 years had a tumor.”
Lines 256-261: It would be helpful to label the four phenotypes with numbers or letters or at least use semicolons to separate them.
Reply. Based on the comment, we added numbers as follows:
“Clinical manifestations are divergent, and thus are classified into four phenotypes: 1) distinctive limbic encephalitis (short-term memory loss, confusion, abnormal behavior, and seizures), 2) limbic dysfunction and multifocal encephalopathy (seizures, psychiatric manifestations, CAs, abnormal movements), 3) limbic encephalopathy preceded by motor deficits, such as weakness, and 4) psychosis with bipolar features [50].”
Line 279: Clarify who “they” are. All patients or the ones with lymphoma?
Reply. We agree with the ambiguous expressions. Accordingly, we changed these sentences as follows:
“The patients received immunotherapies, including IVIg, steroids, mycophenolate mofetil, cyclophosphamide, and rituximab, alone or in combinations. 40% of the patients showed significant improvement or complete resolution of symptoms, while 52% showed stabilization or mild improvement.”
Line 291: “Interestingly, CSF from these patients did not change the density of mGluR2.” In what system was this studied?
Reply. We agree with the criticism. Accordingly, we changed this sentence as follows:
“Interestingly, CSF from these patients did not change the density of mGluR2 in cultures of rat hippocampal neurons.”
Line 299: Same question. In what system?
Reply. We agree with the criticism. Accordingly, we added the information of experimental system.
“Application of serum IgGs obtained from these patients decreased the density of mGluR5 in cultures of rat hippocampal neurons.”
Lines 301-303: two issues of verb-subject agreement. It can be either “encephalitis is” or “encephalitides are”. Likewise, “pure types of CAs” or “pure type of CA”.
Reply. We appreciate the careful check. Accordingly, we changed these sentences as follows.
“Anti-GABAA R Ab- and anti-GABAB R Ab-associated encephalitides are well described [57-60]. However, CAs are rarely observed in patients with anti-GABAA R and anti-GABAB R-associated encephalitis, and the pure type of CA with these autoantibodies is also rare”
Line 308: “Most of patients showed full or partial recoveries to immunotherapies” Most of these patients showed full or partial recovery with immunotherapy.
Reply. Based on the comment, we changed this sentence as follows:
“Most of patients showed full or partial recoveries with immunotherapies.”
Line 318: “not usually associated with CAs.” Change to “but is not usually associated with CA”
Reply. Based on the comment, we changed this sentence as follows:
“, but is not usually associated with CA.”
Line 318: “Half the patients” Half of how many patients? Are these data from reference 59? It would be good to give an idea of how many patients were in the study.
Reply. We agree with the comment. Accordingly, we changed this sentence as follows:
“One study showed that half of the 15 patients had paraneoplastic conditions (lung tumors).”
Line 357: typo, this should be an italicized subtitle
Reply. We appreciate careful check. We changed this as an italicized subtitle.
Line 363: “CAs were described as accompaniment with meningoenchephalomyelitis (40%)” Change to “CA was described as accompanying meningoencephalomyelitis”.
Reply. We agree with the comment. Accordingly, we changed this sentence as follows.
“CAs were described as accompanying meningoencephalomyelitis (40%) [14].”
Lines 390-391: Individual patients have ataxia not ataxias.
Reply. Based on the comment, we changed this sentence as bellows:
“which include pancerebellar ataxia, dysarthria, dysphagia, dysgeusia, oculomotor abnormalities, altered facial sensation, facial nerve palsy, vertigo, pyramidal signs and sensory disorders [15, 68].”
Line 407: “have” is a better term than “develop”, since they don’t develop specific HLA types.
Reply. Based on the comment, we changed this sentence as bellows:
“Patients with PACA characteristically have features suggestive of autoimmune etiologies.”
Lines 409-411: “This HLA type is observed in certain autoimmune diseases (celiac disease, GA, type 1 diabetes mellitus, Stiff-Person syndrome, autoimmune thyroid disease, and autoimmune polyendocrine syndromes), but not in healthy subjects”. This statement would imply that the DQ2 HLA type is never seen in anyone without autoimmune disease.
Reply. We agree with the comment. Accordingly, we changed this sentence as bellows:
“This HLA type is overrepresented in patients with certain autoimmune diseases (celiac disease, GA, type 1 diabetes mellitus, Stiff-Person syndrome, autoimmune thyroid disease, and autoimmune polyendocrine syndromes), given that this HLA marker is found in up to 35% of healthy subjects.”
Lines 416-417: “Consistently, immunohistochemical studies show various staining patterns (60% of the patients)” Staining patterns for what in which system with which antibodies? What does 60% mean in this context, particularly when coupled with the term consistently in the same sentence?
Reply. We agree with the comment. Accordingly, we changed this sentence as follows:
“Consistently, a immunohistochemical study show that cerebellar Abs were present in 60% of the patients with idiopathic sporadic ataxia [16]. Four different staining patterns were observed (cytoplasmic alone, cytoplasmic with processes, nuclear in Purkinje cells, and granular cells) [16]”
Lines 420-421: “However, it should be acknowledged that PACA include seronegative CAs with auto-immune nature”. PACA includes seronegative CAs with auto-immune etiologies.
Reply. We appreciate the comment. We corrected the mistake.
“However, it should be acknowledged that PACA includes seronegative CAs with autoimmune nature.”
Lines 429-431: “Anti-Homer 3, anti-CARP VIII, anti-PKC-γ, and anti-TRIM9/67 Abs are related exclusively to the clinical presentation of CAs.” This is hard to understand at first glance. Perhaps a better way to phrase it would be: The only clinical manifestation associated with anti-Homer 3, anti-CARP VIII, anti-PKC-γ, and anti-TRIM9/67 autoantibodies is cerebellar ataxia.
Reply. We agree with the comment. Accordingly, we changed this sentence as follows:
“The only clinical manifestation associated with anti-Homer 3, anti-CARP VIII, anti-PKC-γ, and anti-TRIM9/67 autoantibodies is CA.”
Line 454: The Figure in the Conclusions section does not look right. Something was lost in how it was put into the manuscript.
Reply. We agree with the comment. Accordingly, we added the following sentence:
“Based on the above review, we highlight two important aspects in the diagnosis of IMCAs. The Figure shows these two factors: “CAs as predominant clinical feature or part of a more diverse presentation” and “Well-characterized or poorly-characterized autoantibodies” (Figure).”
Table 3: Under “nature of autoantigens”
PKC-gamma : remove “is”
Neuronal intermediate filament: change to “cytoskeletal structural component”
Header in column 2: Should be “Frequency of Cerebellar Ataxia”
Reply. We appreciate the careful check. We changed these expressions.
Reviewer 2 Report
I would like to start by congratulating the authors on their paper. Hadjivassiliou, Manto and Mitoma provide us an extensive and comprehensive review on rare immune mediated cerebellar ataxias. The paper is well organized into different sections, according the underlying immune mechanism, and summarized tables. The final figure gives an important perspective on the discussed subjects.
Author Response
We thank the Reviewer for the careful reading and the positive comments.
Reviewer 3 Report
I have read the manuscript with a great interest. This article will pay attention of clinicians towards immune-mediated ataxias during differential diagnosis. The article is well written, describing in detailes the potential causes of some rare cases of ataxia. The ratio of reviews/original articles in References section is appropriate.
Author Response

(The authors gave the same response as above.)
